# Subframe-Level Synchronization in Multi-Camera System Using Time-Calibrated Video

**DOI:** 10.3390/s24216975

**Published:** 2024-10-30

**Authors:** Xiaoshi Zhou, Yanran Dai, Haidong Qin, Shunran Qiu, Xueyang Liu, Yujie Dai, Jing Li, Tao Yang

**Affiliations:** 1National Engineering Laboratory for Integrated Aero-Space-Ground-Ocean Big Data Application Technology, SAIIP, The School of Computer Science, Northwestern Polytechnical University, Xi’an 710129, China; zxs00@mail.nwpu.edu.cn (X.Z.); qhd@mail.nwpu.edu.cn (H.Q.); sranqiu@mail.nwpu.edu.cn (S.Q.); 2The School of Telecommunications Engineering, Xidian University, Xi’an 710071, China; yrdai@stu.xidian.edu.cn (Y.D.); 22011210671@stu.xidian.edu.cn (X.L.); 18010100326@std.xidian.edu.cn (Y.D.); jinglixd@mail.xidian.edu.cn (J.L.)

**Keywords:** video synchronization, time calibration, multi-camera system, subframe synchronization, non-hardware triggered

## Abstract

Achieving precise synchronization is critical for multi-camera systems in various applications. Traditional methods rely on hardware-triggered synchronization, necessitating significant manual effort to connect and adjust synchronization cables, especially with multiple cameras involved. This not only increases labor costs but also restricts scene layout and incurs high setup expenses. To address these challenges, we propose a novel subframe synchronization technique for multi-camera systems that operates without the need for additional hardware triggers. Our approach leverages a time-calibrated video featuring specific markers and a uniformly moving ball to accurately extract the temporal relationship between local and global time systems across cameras. This allows for the calculation of new timestamps and precise frame-level alignment. By employing interpolation algorithms, we further refine synchronization to the subframe level. Experimental results validate the robustness and high temporal precision of our method, demonstrating its adaptability and potential for use in demanding multi-camera setups.

## 1. Introduction

Multi-camera systems significantly enhance the ability to capture and express 3D information compared to single-camera setups. This capability makes them indispensable in advanced visual research fields, including 3D reconstruction [1,2,3,4], multi-view object tracking [5,6], neural radiance fields [7,8,9], and 3D pose estimation [10,11]. The effectiveness of these applications largely depends on the precise synchronization of the cameras involved.

Traditional synchronization methods typically rely on hardware-triggered systems, where cameras are connected via synchronization signals to ensure simultaneous recording. While effective, this approach is often costly and inflexible, particularly in large-scale setups like sports fields or stage performances, where the setup process is both labor intensive and time consuming. As a result, there is growing interest in developing non-hardware-triggered methods that can offer similar synchronization accuracy without the associated complexities and costs.

Non-hardware-triggered synchronization for multi-view cameras was first proposed by Stein [12] and, since then, several techniques have been developed. These include software-triggered synchronization using wireless signals [13,14,15,16,17], traditional feature-based methods relying on geometric analysis [18,19,20,21,22,23,24,25,26,27], and, more recently, approaches based on deep learning [28,29,30]. While these methods can provide satisfactory results under specific conditions, they often have limitations, such as the need for common moving objects across views or stable wireless environments, restricting their general applicability.

In this work, we propose a novel approach to address the synchronization challenge without relying on hardware triggers. Our method introduces a time-calibrated video containing specific markers and a uniformly moving ball to establish a robust global time reference. This enables us to align the local time systems of different cameras and achieve synchronization at the subframe level. The overall framework of our approach is depicted in Figure 1, comprising four main components:**Recording Time-Calibrated Video:** Prior to capturing the target scene, a time-calibrated video is recorded.**Unifying Time Coordinate Systems:** By detecting time-calibrated information across all videos, we derive the relationship between the global time coordinate system and the local time systems of the cameras, enabling synchronization to a unified time reference.**Recomputing Timestamps:** Using the global time information and the original video timestamps, we recalculate the timestamps for all frames, achieving frame-level alignment.**Subframe Level Synchronization:** Due to differences in the startup times between cameras, the aligned video sequences may exhibit a time difference of up to one frame. This can cause noticeable misalignment for fast-moving objects in the scene. However, our method calculates time at a subframe level, allowing us to apply interpolation algorithms to smooth transitions between frames. By interpolating between frames, we can correct these timing discrepancies, eliminating temporal errors in fast-paced scenes and ensuring a more accurate visual alignment across camera views.

Our method imposes no specific requirements on the cameras, recording methods, or captured objects, making it highly versatile and adaptable. The only prerequisite is that the playback device for the time-calibrated video should have a refresh rate at least equal to the frame rate of the video, a condition easily met by most modern devices.

In summary, our contributions include the following:A novel non-hardware-triggered synchronization method for multi-camera systems, utilizing a time-calibrated video as a global time reference.Development of a robust algorithm for detecting time-calibration information, enabling stable synchronization with a minimal setup.Achievement of subframe precision in synchronization, enhancing the temporal accuracy beyond the frame level through interpolation algorithms.

## 2. Related Work

In this section, we overview three major categories of non-hardware triggered video time-alignment algorithms, including software-triggered time-alignment algorithms, algorithms based on various features for time alignment, and algorithms utilizing deep learning for time alignment. Additionally, we introduce the current research on video-frame interpolation algorithms using deep learning.

### 2.1. Software-Triggered Synchronization

In distributed systems, achieving precise time synchronization typically involves using a set of physical synchronization devices. These devices synchronize the operations of all devices to occur at almost the same moment by triggering them using physical signals. Some research has explored replacing wired connections with wireless ones to trigger simultaneous operations across all devices. Ahrenberg et al. [13] and Litos et al. [16] both utilize the Network Time Protocol (NTP) to align device clocks, enabling simultaneous operations under the same wireless signal source after alignment. Litos et al. [16] designed a variant of NTP that considers the delay between signal reception and actual execution, further enhancing the accuracy of wireless signal synchronization. In contrast to previous methods requiring operations on camera sensors or utilizing smartphone hardware image streams, Bortolon et al. [15] proposed a completely software-based wireless communication method. This method achieves both device time synchronization and calibration, along with 3D skeleton and dynamic object reconstruction.

In situations where the wireless network conditions are favorable, the methods mentioned above exhibit good time synchronization accuracy. However, they may fail in scenarios where devices are too far apart or experience interference. Additionally, these methods impose certain requirements on the devices used. Currently, devices capable of implementing these methods are mainly similar to smartphones, as they can install relevant software, connect to networks, and transmit wireless signals.

### 2.2. Traditional Feature-Based Synchronization

Due to the establishment of effective geometric correspondences in multi-view scenarios, feature-based video time-alignment methods rely on the features of images combined with the fundamental matrix. Most studies use feature point detection in images without specific constraints, although some research utilizes specialized feature points for particular research contexts. For instance, Wang et al. [26] use key points of the human body, while Sinha and Pollefeys [24] employ contour points of moving objects. For frame-level synchronization, most methods match detected dynamic feature points and establish epipolar geometry relationships to obtain synchronization. To achieve subframe-level synchronization precision, methods such as those proposed by [18,19,22,23] establish temporal constraint relationships and iteratively optimize to obtain the required subframe-level registration coefficients. Tresadern and Reid [25] assume linear changes in dynamic feature points and utilize interpolation to achieve subframe precision. In terms of video-frame synchronization algorithms, Imre and Hilton [20] use graphs for synchronization fusion, while Yan and Pollefeys [27] directly derive synchronization results using the correlation of spatio-temporal interest point distributions, bypassing feature matching. Breaking away from previous methods that only use two viewpoints to establish constraints, Lei and Yang [21] utilize three viewpoints to establish geometric constraints, making the obtained synchronization results more robust.

In addition to using image features for video time synchronization, a multi-view video time synchronization system has been constructed based on features extracted from audio captured by cameras. Leveraging the hardware characteristics of certain cameras, such as rolling shutter cameras, can produce unique features on images under rapidly changing lighting conditions, such as changes in light intensity edges. Bradley et al. [31] and Smid and Matas [32] propose high-precision video time synchronization systems based on these unique features extracted from the audio captured by cameras.

Traditionally, feature-based synchronization methods align sequences based on feature trajectories. These methods can achieve high alignment accuracy when good features are detected, but they come with various requirements and limitations. For methods relying on image features related to dynamic objects, it is necessary to have common dynamic objects across multiple views. Both audio and image feature-based methods are sensitive to noise interference, which can result in poor alignment performance in the presence of noise. Moreover, methods utilizing image features generated by hardware conditions impose direct requirements on devices and shooting environments. Additionally, the resulting image quality may not meet the requirements of some applications due to the presence of distinct boundaries caused by changes in light intensity.

### 2.3. Deep Learning-Based Synchronization

Similar to traditional methods, video time synchronization based on deep learning also involves learning features from each frame of the video and then using algorithms such as Dynamic Time Warping (DTW) for time alignment. Haresh et al. [30] utilized Contrastive-Inter-Device Matching (Contrastive-IDM) to regularize the extracted features, mapping points to corresponding embedding spaces based on the distance between frames, and then performed time alignment using Soft-DTW. Boizard et al. [28] designed a model for stereo cameras to compute alignment scores between frames and then utilized these scores to calculate the average delay for aligning the two lenses of the stereo camera. Fakhfour et al. [29] used both global and local features together for feature learning, and then performed sequence alignment using the Derivative Dynamic Time Warping (DDTW) algorithm. Additionally, some deep learning algorithms related to video sequence alignment seem to be focused on representation learning tasks [33,34,35], aimed at aligning time sequences of two or more videos with similar behaviors. In cases where there is not much difference in viewpoints and the dynamic objects being filmed are the same, this deep learning approach can also be used for the time synchronization of multi-view videos.

The method of utilizing deep learning frameworks for feature extraction is similar to traditional methods in that it requires high-quality features of the filmed objects. Additionally, when dealing with long sequences or high-resolution video sequences, deep learning consumes significant computational resources.

### 2.4. Deep Learning-Based Video Interpolation

Video-frame interpolation algorithms are typically employed to insert additional frames between existing frames in a video, thereby enhancing the overall smoothness and viewing experience. These algorithms fundamentally involve inserting extra frames between existing ones. Currently, major video-frame interpolation algorithms can be broadly categorized into two types: those directly leveraging the learning capability of neural network frameworks for interpolation and those combining motion modeling with deep learning for interpolation. Methods that directly utilize neural network frameworks to learn the information between two frames are relatively straightforward. However, due to the lack of motion modeling, the interpolated frames may not correspond well to the regions between input frames, resulting in blurry or ghosting artifacts (Lee et al. [36]). On the other hand, motion-aware methods typically utilize optical flow to model the motion between two frames and predict intermediate frames based on this modeling. Most of these optical flow-based methods [37,38,39,40,41] also incorporate contextual feature information to assist in generating intermediate frames, significantly improving the effectiveness of video-frame interpolation and achieving remarkable results. Since our video synchronization algorithm rewrites the timestamps of synchronized videos, we can utilize an advanced video-frame interpolation algorithm to further generate video sequences with higher synchronization accuracy.

## 3. Method

The non-hardware-triggered multi-camera system synchronization method proposed by us mainly consists of four modules. Figure 2 illustrates the complete process of our method. Our time-calibrated video acts as a reference clock. Before shooting the target object, we play this video using a playback device with a refresh rate higher than the video frame rate. All cameras then record the video content stably for several seconds. After shooting the target, we use the embedded time-calibrated information from the videos recorded by all cameras to establish the relationship between the global time system and the local time systems of the cameras. By using these relationship, we recalculate the timestamps for all videos and rewrite the timestamps. Given the potential large resolution and length of the videos, we employ the NVIDIA hardware codec [42]. Ultimately, for subframe-level alignment, we introduce the frame interpolation algorithm to interpolate the frame-level alignment output using the rewritten timestamps.

This section provides a detailed explanation of the key components of our method. The required input for this method consists of video content Videos$=\{V_{1},V_{2},\ldots ,V_{N}\}$ along with time information exported by *N* cameras. After rewriting the timestamps, we obtain the frame-level alignment, referred to as FrameAlignedVideos. Following frame interpolation, we achieve the subframe-level alignment, denoted as TimeAlignedVideos.

### 3.1. Time-Calibrated Video Design

To synchronize multiple camera systems, we design a time-calibrated video that acts as a global timing reference, essential for aligning the independent time systems of each camera. Each camera records this video before capturing the main content, ensuring synchronized video acquisition.

#### 3.1.1. Key Design Elements

To enhance the synchronization effect, we base our time-calibrated video on three essential features:**Ease of Detection:** The video should facilitate easy extraction of time information from each frame. *ArUco markers* [43] are widely used in computer vision due to their high detection accuracy, even under challenging conditions such as rotation, scaling, or varying light. Their low computational demands make them suitable for real-time processing. Moreover, each marker has a unique ID, preventing repetition. These qualities establish *ArUco markers* as the ideal choice for our time-calibrated video, facilitating reliable synchronization.**Stability:** To minimize errors during rapid scene changes that may lead to frame discrepancies, we incorporate blank frames into the video design. These blank frames help reduce artifacts during transitions, ensuring more stable time detection.**High Precision:** To achieve millisecond-level precision, we incorporate a uniformly moving ball into the video. The ball’s smooth, continuous motion enables high-precision timing, contributing to finer granularity. Its position can be easily tracked using traditional computer vision methods, making it suitable for real-time synchronization.

#### 3.1.2. Video Structure

The final time-calibrated video, shown in Figure 2, acts as a global clock and consists of two main parts. The left part provides second-level time information, while the right part offers millisecond-level details. If the video has *f* frames per second, the left side maintains consistent content for f−2 frames, with the last two frames left blank to avoid visual errors during time transitions. The right side features a ball moving uniformly from the bottom-left to the top-right corner, representing the passage of time from 0 to 1000 ms.

We employ a 7×7 *ArUco marker* configuration, which allows for up to 1000 unique markers. However, the configuration can be adjusted based on the required video duration. These markers serve three specific purposes:**Loop markers** (red in Figure 2a) indicate cycles.**Group markers** (blue in Figure 2a) provide group-level information.**Positioning markers** (green in Figure 2a) help determine the ball’s exact position for millisecond-level precision.

We designed groups of 8 markers without overlap between them. This design ensures that, even if the time-calibrated video is recorded for a short period, each camera can reliably capture the global time information. Loop markers can be detected multiple times within seconds, and the loop typically remains unchanged. If this consistency is maintained, previously detected loop information can be reused for calculation. In contrast, group markers represent time intervals of just 1 s, necessitating a higher level of detection accuracy.

Positioning markers also require precision, particularly in aligning millisecond-level time data. Once the second-level time is determined, the millisecond information can be used to adjust the second-level timing according to each camera’s local time. Section 4.2.1 provides a detailed validation of the rationale behind this design.

#### 3.1.3. Time Calculation

The second-level time, $T_{s}$, is calculated using the following formula:(1)$$T_{s}=\left(G\times l+g\right)\left(s\right)$$
where *G* represents the number of groups, *l* is the cycle number, and *g* is the group number for the current frame.

The right side of the time-calibrated video, shown in Figure 3a, includes four positioning markers located within green boxes and a moving ball. These markers form a large square, with the lower-left corner acting as the origin for a coordinate system. By scaling this system, we can determine the UV coordinates of the ball for the current frame. Since the ball’s starting and ending coordinates are predetermined, we can compute the millisecond time using the following equation:(2)$$T_{ms}=\left(\frac{u_{b}-u_{s}}{u_{e}-u_{s}}+\frac{v_{b}-v_{s}}{v_{e}-v_{s}}\right)\times \frac{1000\times (f-1)}{f\times 2}\left(ms\right)$$
where *f* is the frame rate (in frames per second), and $\left(u_{s},v_{s}\right)$, $\left(u_{e},v_{e}\right)$, and $\left(u_{b},v_{b}\right)$ are the starting, ending, and current frame coordinates of the ball, respectively.

### 3.2. Time System Alignment

After the recording is complete, the video sequences Videos exported by the cameras undergo a two-step process. The first step involves aligning the local time systems of all Videos with the global time system. In the second step, the timestamps for each frame within the global time system are recalculated and rewritten.

For the input video $V_{i},i\in N$, we want to establish a correlation between the time $RawTimes\left(V_{i}\right)=t_{r_{i1}},t_{r_{i2}},\ldots ,t_{r_{in_{i}}}$ (ms) under $Cam_{i}$ time system and the time $PublicTimes\left(V_{i}\right)=t_{p_{i1}},t_{p_{i2}},\ldots ,t_{p_{in_{i}}}$ (ms) under global system. The entire algorithm process is depicted in Figure 4a. The algorithm first decodes $V_{i}$ and locates the segment of the video that starts recording the public time information by detecting the group markers in the time-calibrated video. To enhance stability, the public time information is not immediately associated with the $V_{i}$ time system. The algorithm will start counting the frames. Upon the next group change, the resulting count will be the new fps. It is then compared with real fps. If the difference is less than 1, it is considered a stable detection time, and the current group number $g_{1}$ is recorded. Subsequently, an attempt is made to detect the loop marker. If not detected (i.e., $l_{1}=\emptyset$), the real time of the current frame $t_{1}$ is recorded, and the algorithm continues to read the subsequent frames. When the loop marker is detected, the current cycle number $l_{2}$ and real time $t_{2}$ are recorded. The group number of the current frame $g_{2}$ can then be calculated as $g_{2}=\left(g_{1}+\mathrm{floor}\left(t_{2}-t_{1}\right)\right)\%G$. If the cycle marker is detected, then $l_{2}=l_{1}$, $g_{2}=g_{1}$, and $t_{2}=t_{1}$.

Through the above process, the second information $T_{s_{i}}$ in the global time system can be obtained by formula (Equation 1), and then the detection of millisecond information begins. Since deformation may occur during the shooting process, all four positioning markers must be detected. Then, the deformed rectangle is transformed into an orthogonal rectangle through orthogonal projection using the corner information of positioning markers. The square coordinate system in which the ball is moving can then be located, and the millisecond information $T_{ms_{i}}$ can be obtained using formula (Equation 2). This process may also fail, so our algorithm takes this into account and continues to detect the next frames when it fails. When the information of the four positioning markers can be completely detected, the real time of the current frame $t_{3}$ is recorded. Then, the group information is updated by $g_{3}=\left(g_{1}+\mathrm{floor}\left(t_{3}-t_{1}\right)\right)\%G$ and $T_{s_{i}}$ also needs updating. The millisecond information $T_{ms_{i}}$ is still obtained using formula (Equation 2). Assuming the current frame index is *j*, then, we can obtain $t_{p_{ij}}=T_{s_{i}}\times 1000+T_{ms_{i}}$. We obtain the association as follows:(3)$$t_{p_{i(j+m)}}=t_{p_{ij}}+\left(t_{r_{i(j+m)}}-t_{r_{ij}}\right),m\in \left(0,n_{i}-j\right)$$
where *m* represents the frame number after the aligned frame *j*, $n_{i}$ represents the frame numbers of $V_{i}$, and $t_{r}$ denotes the local millisecond time, which is calculated based on the timestamp embedded within the video.

After obtaining the association between the corresponding local time systems of all videos and the global time system as (Equation 3), we will take the moment with the maximum calculated global time, that is $Max(t_{p_{ij}})$, as the latest alignment moment. All videos will rewrite the timestamps from this moment to output FrameAlignedVideos, as shown in Figure 4b.

### 3.3. Video-Frame Interpolation

To enhance the accuracy and usability of frame alignment in videos, we introduce a video interpolation algorithm. After the previous step, where the transformation relationship between each video frame and the global time system is established, the alignment time for all cameras is marked by the yellow dotted box in Figure 4b. Following the alignment, we recalculate the timestamp for each camera, starting from zero at the alignment moment.

The new timestamps are generated under the global time system based on the time information from the existing frames and video timescale. Using these recalculated timestamps, we output a sequence of aligned videos, referred to as FrameAlignedVideos.

Building on FrameAlignedVideos, and using a given frame rate $f_{given}$, we perform interpolation across the video timeline. The time sequence is defined as follows:$$\left[0\;\mathrm{ms},\frac{1}{f_{given}}\times 1000\;\mathrm{ms},\frac{2}{f_{given}}\times 1000\;\mathrm{ms},\ldots \right]$$

This interpolation process ensures that all videos are adjusted to a common time sequence, resulting in TimeAlignedVideos, where the frames across different cameras are temporally synchronized according to the global time system.

### 3.4. Real-Time Feasibility Discussion

Although the current method is not real-time due to resource limitations and implementation complexity, we can analyze its potential for achieving real-time performance. In a multi-camera system, it is essential to first capture time-calibrated video before the main recording task. In real-time scenarios, where synchronized video data are required after recording starts, we propose using a multi-GPU server to manage video-frame acquisition and synchronization. Each camera is assigned its own GPU resource, while a shared GPU memory space is set up for coordination.

**Global Alignment Time Calculation.** This stage, which calculates the global alignment time, occurs during preprocessing and does not need strict real-time performance. As the cameras record time-calibrated video, real-time encoding and decoding can be achieved using modern codecs, allowing us to extract both frame images and local time information in real time. Once the global time is calculated, it is written into the shared GPU memory along with the current frame’s timestamp. When the last camera completes its write operation, the global time is set as the latest alignment time (shown by the yellow dashed box in Figure 4b). The algorithm’s complexity is mainly determined by the marker detection for each frame. If the time of capturing time-calibrated video is *t* seconds (where t>3 seconds) and the frame rate is *f*, the maximum complexity is O(t×f). In Section 4.2.2, we perform a brief analysis of the time performance for this step.

**Frame Alignment Stage.** At this stage, real-time performance becomes critical. The latest alignment time is stored in the shared GPU memory, accessible at any time by the GPU resources managing the video streams. Once the global time is calculated, each camera’s GPU continuously recomputes based on the global and local time information (as detailed in Section 3.2). This process involves only simple arithmetic operations, which take minimal time. The recalculated time then replaces the original timestamp in the video frames. Once the final alignment time is set, an alignment signal is sent to all video streams, functioning similarly to a hardware synchronization signal. This allows the output of synchronized video streams, with operations performed entirely on the GPU. The main time cost in this stage is video encoding and decoding, which current technology can handle effectively.

**Frame Interpolation Stage.** To enhance video alignment precision, we introduce a frame interpolation algorithm at this stage. For higher alignment accuracy, we recommend using a deep learning-based approach. While traditional methods offer good real-time performance, deep neural networks deliver better quality in aligned video sequences. Current interpolation algorithms, such as RIFE [38], can achieve near real-time performance, interpolating 720p video at 30 fps on an NVIDIA 2080Ti GPU, although their quality may not match that of non-real-time algorithms. This presents an opportunity for future improvement, focusing on developing an interpolation algorithm that balances real-time performance with high quality. Once the alignment signal is received, the interpolation process begins. This step involves extracting images from two adjacent frames and performing frame interpolation to produce the aligned video stream, as detailed in Section 3.3, along with new timestamp information.

### 3.5. Potential Issues and Solutions

**Environmental Challenges**: Since the time-calibrated video must be displayed on electronic devices and recorded by all cameras, issues may arise due to lighting conditions or obstructions that prevent accurate detection of time information. To mitigate this, if the cameras are movable, it is recommended to first record the time-calibration video in a well-controlled environment. Afterward, the cameras can be repositioned while remaining powered on to capture the target scene. In situations where camera movement is not feasible, it is advisable to monitor the camera feed during the time-calibration recording and make artificial lighting adjustments as needed to ensure optimal capture quality.

**Equipment Limitations**: Cameras with lower resolutions may struggle to detect time markers accurately. In such cases, smaller ArUco markers, such as 5×5 grids, can be used, and the number of markers per group can be reduced to three to enhance detection. While this approach may sacrifice some stability, longer recording times can help compensate for the loss of precision. However, this method is unsuitable for devices lacking video encoding capabilities and those that cannot maintain stable frame rates (fps). For such devices, both time synchronization and recalculating time information require local video timestamps. If a device lacks encoding but maintains a stable fps, the time information from the calibration video can be used to determine the fps during pre-calibration, allowing for the estimation of local timestamps.

**Interpolation Limitations**: To achieve frame-level alignment, our method employs interpolation algorithms. However, these algorithms typically perform better in linear motion scenes, and the deep learning networks trained for this purpose may not fully address real-world interpolation needs, limiting their applicability to certain visual tasks. Moreover, for fast-moving objects, the precision of our algorithm is restricted to subframe levels, potentially resulting in noticeable errors in such scenarios. Additionally, our method achieves only subframe-level precision, meaning it does not align to the millisecond or sub-millisecond level. Consequently, this inherent margin of error limits its applicability in scenarios involving very-fast-moving objects. Currently, our method is most effective for scenes characterized by relatively uniform and moderately paced motion.

## 4. Experiments

### 4.1. Experimental Settings

Our experiments are divided into four parts. The first experiment analyzes the performance of our algorithm, focusing on how the design of multiple markers forming a group contributes to the algorithm’s stability, as well as testing the algorithm’s time performance. The second experiment utilizes a hardware-synchronized dataset to assess the accuracy of our method. In the third experiment, we compare the synchronization results of our algorithm with those of two other synchronization algorithms [33,34], using data captured from cameras without hardware synchronization to analyze the stability of our algorithm. The fourth experiment evaluates the effectiveness of millisecond-level information by employing a frame interpolation algorithm [41] in conjunction with the state-of-the-art scene reconstruction algorithm, 3D Gaussian Splatting [44]. Each of these experiments involves different cameras, demonstrating the versatility of our algorithm.

The experiments are conducted on a system powered by an AMD^®^ Ryzen 5 4600H with Radeon Graphics (12 cores), 16 GB of RAM, and an NVIDIA RTX 1650 GPU. The operating system used is Ubuntu 20.04 LTS, and the implementation is carried out in C++ using OpenCV 4.6.0 [45], CUDA 11.7 [46], and the NVIDIA VIDEO CODEC SDK 10.0 [42] for GPU acceleration.

#### 4.1.1. Datasets

While some public datasets are available, they do not meet our specific requirements for time-calibrated videos. Therefore, we created a custom dataset for more precise experimental validation. We designed three sets of shooting scenarios as datasets for our experiments:For the first experiment, data were captured using a hardware-synchronized multi-camera system (*Femto Bolt* [47]) to film indoor ball-playing movements. Frames were extracted at intervals from the synchronized data to create unsynchronized data.The second experiment involved data captured using a multi-camera system without hardware synchronization (*Hikvision* [48]), focusing on pouring scenes and scenes of a ball falling.The third experiment utilized a panoramic multi-camera system (*Insta*-×3 [49]) to film indoor ball-playing movements.

The datasets for these three experiments are illustrated in Figure 5. All shooting took place indoors, within a cylindrical scene enclosed by a custom calibration cloth.

#### 4.1.2. Evaluation

In the second experiment, we utilize video sequences captured by a hardware-synchronized multi-camera system as our experimental data. The objective is to validate the error range of our synchronization algorithm. After obtaining the synchronization results, we calculate the error based on the time information derived from our algorithm (including millisecond-level data) and the time information provided by the hardware synchronization system, which serves as the ground truth. Since our method primarily requires identifying alignment moments and recalculating time information from the original video, we measure our error by assessing the misalignment accuracy between cameras at the aligned frames. For aligned frames, the predicted times for cameras *i* and *j* are denoted as $t_{p_{i}}$ and $t_{p_{j}}$, respectively, while the recomputed true times are denoted as $t_{r_{i}}$ and $t_{r_{j}}$. The error between cameras *i* and *j* is defined as follows:(4)$$\Delta t_{ij}=\left|\left(t_{pi}-t_{pj}\right)-\left(t_{ri}-t_{rj}\right)\right|$$

In the third experiment, we use video sequences captured by a non-synchronized multi-camera system. Here, we compare our synchronization algorithm with two deep learning algorithms [33,34] to validate the stability of our method. We showcase visually noticeable transition moments for a subjective evaluation of synchronization and quantitatively assess synchronization using the epipolar geometry relationship of a falling ball. Let a point in space have projections $p_{1}$ in camera 1 and $p_{2}$ in camera 2. The epipolar line formed by camera 1 about the point in camera 2 is denoted as $l_{2}$. The pixel distance from $p_{2}$ to $l_{2}$ is measured as dist. The frame with the minimum dist is considered as the alignment frame of camera 2 relative to camera 1. If the alignment result is correct, the dist of the alignment frame compared to neighboring frames should be the smallest.
(5)$$dist=d(p_{2},l_{2})$$

In the fourth experiment, we utilize video sequences captured by a non-synchronized panoramic multi-camera system. The purpose of this experiment is to validate the effectiveness of the millisecond-level information obtained from our algorithm. To achieve this, we employ the 3D Gaussian Splatting algorithm [44] to reconstruct FrameAlignedVideos and TimeAlignedVideos and render new perspective images for comparison. In addition to visual assessment, we use a Laplacian operator to quantitatively evaluate the quality of the rendered images based on their Laplacian gradients. A larger Laplacian gradient indicates higher image quality.

### 4.2. Algorithm Performance Analysis

#### 4.2.1. Stability Analysis

In this section, we aim to validate the rationale behind the design of group markers, specifically whether the use of multiple markers contributes to detection stability. We utilize data captured by two unsynchronized cameras (as illustrated in Figure 5b), which have a resolution of 3840×2160 and a frame rate of 20 fps. These cameras are primarily for monitoring purposes and possess relatively average image quality, making them suitable for stability analysis—particularly to assess whether group markers can effectively capture group information under suboptimal conditions.

We calculate the number of group markers detected within each frame for both cameras from the beginning of the time-calibrated video recording. The final results are presented in a pie chart in Figure 6. It can be observed that the marker detection numbers for camera 1 are somewhat unstable, with a maximum of eight markers detected (indicating that all markers in the group were recognized) and a minimum of zero markers. Notably, 9.53% of the frames contained only one detected marker. In contrast, camera 2 exhibits more consistent results, with the majority of frames detecting all markers; however, some frames still showed no markers detected, while a few displayed either one or seven markers.

Based on these experimental results, we conclude that, in conditions where image quality cannot be guaranteed, the design of multiple markers per group ensures that, as long as at least one marker is detected, the group can still be recognized. Therefore, incorporating multiple markers per group significantly improves the detection stability.

#### 4.2.2. Time Performance Analysis

This section analyzes the system’s time performance to validate its real-time capabilities. While the requirements for real-time performance during the alignment step are not stringent, the processing time for each frame should be kept within reasonable limits. Excessive processing times could lead to the accumulation of large volumes of captured data and increased storage consumption.

We utilize data captured by four synchronized cameras (the example illustrated in Figure 5a), with a resolution of 1920×1080 and a frame rate of 30 fps. We divide the processing time into four components: (1) detection time for ArUco markers, (2) time taken to detect loop information, (3) time for detecting group information, and (4) time for acquiring millisecond timestamps, the latter of which is currently implemented using the Hough transform. Both the detection of ArUco markers and the Hough transform utilize the OpenCV library, which can later be implemented on the GPU for enhanced real-time performance.

Table 1 presents the final experimental results. The findings indicate that the primary time-consuming steps are the detection of ArUco markers ($t_{aruco}$) and the detection of millisecond timestamps ($t_{ms}$). In contrast, the detection of loop information ($t_{loop}$) and group information ($t_{group}$) consume negligible time. Overall, the frame detection time is maintained within 100 ms; while this is not classified as real-time, the performance is acceptable for our purposes.

Additionally, based on the global time calculation results, we recorded time statistics for the hardware encoding and decoding components. The average decoding time per frame is $1.851\times 10^{-3}$ ms, while the average encoding time per frame is $1.297\times 10^{-3}$ ms.

### 4.3. System Error Validation

The hardware synchronous multi-camera system uses four cameras with 30 fps (the example illustrated in Figure 5a). We performed a frame offset operation on the original data to generate the input video, ensuring that the error between aligned perspectives is within (1000/30×4) ms. Comparing the aligned true timestamps (we assume that the hardware synchronization equipment achieves perfect synchronization) with the calculated timestamps for each camera using formula (Equation 4), the maximum error is 17 ms and the average error is less than 10 ms, as shown in Table 2.

The main sources of error in our method are primarily two-fold: (1) The inherent time interval of the time-calibration video itself. In this experiment, a time-calibration video with a frame rate of 120 fps was used, resulting in an error of (1000/120) ms. (2) When detecting the motion of the ball, motion blur and detection errors can lead to a maximum error of (1000/120)×2 ms. This means that one camera captures the ball with a ghosting effect appearing in front of its actual position, while another camera captures the ghosting effect appearing behind the actual position of the ball. Therefore, theoretically, there could be an error of (1000/120)×3 ms. To enhance precision, increasing the frame rate of the calibration video could be considered. However, this places higher demands on the camera’s performance to prevent the occurrence of significant artifacts. The overall implementation results, as shown in Table 2, indicate that the maximum error observed is 17 ms. The theoretical analysis predicted a maximum error of 25 ms. Therefore, the overall performance shows an error of less than half of the theoretical value, which aligns with the expected outcome.

### 4.4. Comparison with Existing Methods

We use two non-synchronized cameras for data capture (the example illustrated in Figure 5b), with a frame rate of 20 fps. Then, we compare video-alignment results with two relatively classical deep learning algorithms [33,34]. These two deep learning methods are trained on a scene beforehand to obtain pre-trained models. We select their pre-trained models for pouring scenes and film pouring actions accordingly. To improve the accuracy of the predictions made by the depth methods, we cropped and resized our data using their training data as a reference. Due to the poor prediction results of depth networks on frames with excessive misalignment, the input videos for the depth methods are manually offset by 35 frames after alignment.

Comparing the alignment results as illustrated in Figure 7, in the pouring scene, within camera 1, the water transition from not pouring to pouring is observable between adjacent frames. Likewise, the alignment result from our algorithm for camera 2 also illustrates the water pouring between adjacent frames. However, the CARL [33] algorithm already depicts a significant amount of water poured out, whereas the TCC [34] algorithm indicates the water not yet poured out. For the falling ball scenario, we project the epipolar line $l_{2}$, corresponding to the ball from Cam 1, onto the image from Cam 2 for visual reference. As shown, in our method, the epipolar line $l_{2}$ in two consecutive frames closely aligns with the position of the ball in Cam 2, indicating good synchronization between the two cameras. However, in the CARL method [33], the ball that should appear in Cam 2 never shows up, indicating poor alignment. While the TCC method [34] does detect the ball in Cam 2, it appears far from the epipolar line $l_{2}$, failing to meet the alignment requirements.

Using the epipolar geometry relationship, we select a frame from camera 1 as the target frame and use the center of the falling ball as the target point. We then calculate the dist for the aligned frame of camera 2 using our algorithm and the two other algorithms according to formula (Equation 5). The results are shown in Table 3 (- indicates that the dist cannot be calculated). The frame ID represents the corresponding frames of camera 2 under the alignment results of the the three methods for the *i*th frame of camera 1. From our results, it can be observed that the dist for the aligned frame is the smallest (as shown in Figure 7), meeting the alignment requirement. However, the compared methods do not meet this alignment requirement.

From the experimental results, it is evident that our method obviates the need for pre-training operations and boasts excellent alignment stability and precision. Furthermore, our alignment capability is robust for videos with substantial temporal misalignments, whereas the competing methods often underperform in such scenarios. Consequently, during the experiment, we manually established a narrow misalignment range to ensure that the comparative methods could achieve reasonable alignment results.

### 4.5. Application in Advanced Multi-Perspective Requirement Method

We capture indoor ball playing scenes using 18 panoramic cameras (the example illustrated in Figure 5c), with a resolution of 8192×4096 and a frame rate of 30 fps, and apply the FILM [41] interpolation algorithm. To validate the effectiveness of the millisecond information obtained by our algorithm, we employ the state-of-the-art scene reconstruction algorithm, 3D Gaussian Splatting, to reconstruct the interpolated and non-interpolated frames. We divide the panoramic view into 6 perspectives for calibration and obtain calibration results for 68 perspectives. During the training process of 3D Gaussian Splatting, all data use the same colmap [50,51] calibration results, and the same initial point cloud is used.

As seen in Figure 8, the clarity of the ball in the raw footage without frame interpolation is noticeably lower compared to the interpolated version. Additionally, the raw footage exhibits more prominent artifacts. Regarding facial features, the contours of the person’s face in the interpolated frames are significantly clearer, indicating an improvement in visual quality post-interpolation. Aside from errors inherent to the interpolation algorithm itself (such as bad interpolated results or misalignment with expectations), overall, the reconstruction effect after interpolation is better than before interpolation.

Figure 9 shows box plots of the gradient sums calculated using the Laplacian operator for 100 rendered perspectives under non-interpolated and interpolated conditions. Due to the inherent instability of frame interpolation, the visual quality of the interpolated frames is sometimes lower than the original frames. However, when the interpolation works well, the peak visual quality is even better than that of the raw footage. From both the mean and median values, the overall quality after interpolation is superior, which verifies the effectiveness of the millisecond-level time calculation. This also highlights that current interpolation algorithms can be unstable in certain scenarios, an issue that requires further investigation and improvement in future work.

To showcase the method’s effectiveness, a Appendix A titled “Demonstration of Synchronization Effects Based on Time-Calibrated Video Synchronization Algorithm: Results of Panoramic Camera Synchronization in a 6DOF System”. has been provided. This video demonstrates the performance of our time synchronization method applied within a 6DoF system [52], showcasing different stages in the data capture and synchronization process. It sequentially presents the original data footage, the pre-synchronization state, the results after synchronization of the raw video, and, finally, the output achieved through frame interpolation to enhance alignment precision. These visualizations underscore the effectiveness of the proposed approach in aligning multi-perspective video streams.

## 5. Conclusions

We present a novel synchronization method for multi-camera systems that does not rely on hardware triggers, using a time-calibrated video as the core element. This video consists of two key components: a grid marker pattern that provides second-level timing and a moving ball that ensures millisecond-level precision. By leveraging this global time reference, we align the time coordinate systems of all cameras, leading to the generation of synchronized videos with subframe accuracy through interpolation algorithms.

The proposed method is validated using a hardware-synchronized multi-camera system, where we assess its error margins and alignment performance. Furthermore, comparisons with existing deep learning-based synchronization algorithms highlight the stability and precision of our approach. To demonstrate the practical effectiveness of the millisecond timing information, we apply advanced reconstruction algorithms, showing improved rendering quality in interpolated frames.

Our method is versatile, requiring only the recording of a time-calibrated video, and is compatible with various camera types and scene contents. In certain scenarios, hardware-triggered synchronization may not be economically or technically feasible. Our approach offers a flexible and cost-effective solution when hardware-based synchronization is unavailable. Additionally, the potential for real-time synchronization is achievable with appropriate video encoding and decoding hardware support, making it a strong candidate for real-world applications with stringent synchronization demands.

## 6. Future Work

In future work, we aim to enhance the real-time performance of the proposed approach. Currently, our method is best suited for environments where cameras can capture clear time-calibration video content, and where devices possess robust video encoding capabilities and high-quality image capture.

To improve stability, we plan to integrate illumination compensation algorithms to enhance the robustness of our approach in varied lighting conditions. Furthermore, we will explore enhancements to the interpolation algorithms to better utilize the millisecond-level time information generated by our method. Specifically, our objectives include developing more motion-accurate interpolation techniques and investigating the application of non-linear interpolation algorithms to improve the performance in dynamic scenes.

## Figures and Tables

**Figure 1 sensors-24-06975-f001:**
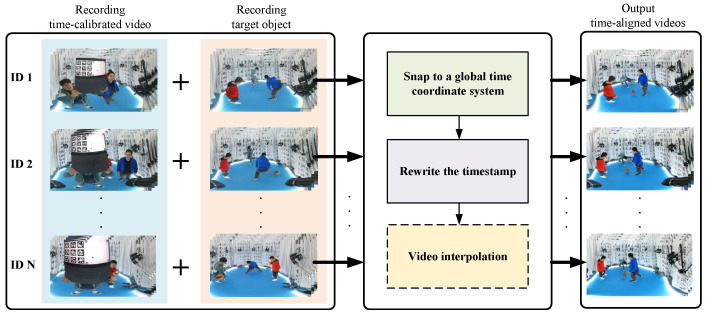
Our proposed framework for non-hardware-triggered video sequence time synchronization based on time-calibrated videos is depicted. The input comprises two main components: a sequence of time-calibrated video captured and the filmed target object (on the left). The algorithm for handling non-aligned videos is delineated into three steps. Initially, unification of the time axes of all videos is performed. Subsequently, leveraging the obtained unified time information and original video timestamps, recalculating and rewriting timestamps are executed, and, if required, employing video interpolation algorithms based on the new timestamps is conducted. Ultimately, aligned videos are obtained.

**Figure 2 sensors-24-06975-f002:**
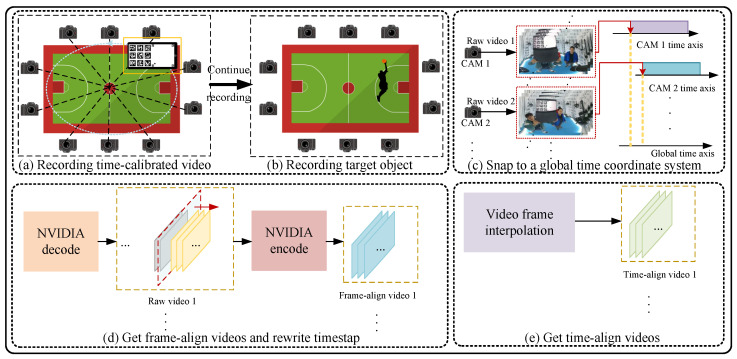
The complete process of non-hardware-triggered multi-camera synchronization based on time-calibrated video can be divided into five main steps: (**a**) After setting up the multi-camera system (assuming it is set up in a basketball court), each camera records a time-calibrated video playing on the other device. Each camera records stably for several seconds and, after recording, the cameras remain powered on. (**b**) After recording the time-calibrated video, the recording of the target scene continues. (**c**) After capturing the scene, the corresponding videos from all cameras are exported. The content of the time-calibrated video is analyzed, and the relationship between each video and the global time systems is extracted. (**d**) Using the information obtained in the previous step and NVIDIA hardware encoding/decoding, the videos with recalculated timestamps are outputted for frame alignment. (**e**) Finally, video-frame interpolation algorithm is utilized to further enhance the accuracy of alignment.

**Figure 3 sensors-24-06975-f003:**
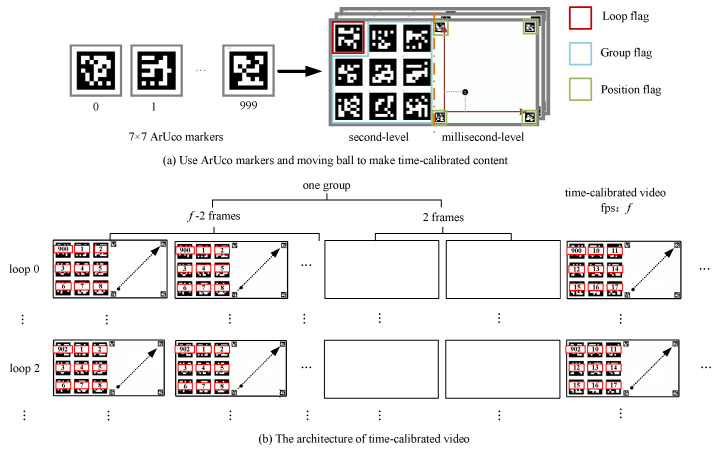
Design of the time-calibrated video. (**a**) We employ a 3×3 marker composed of ArUco markers and a ball with 4 positioning ArUco markers to constitute the content of a frame in the time-calibrated video. The left side represents the second level of time information and the right side represents the millisecond level of time information. (**b**) Assuming the fps of video is *f*, each second of video comprises (f−2) identical nine markers and a small ball moving from the bottom left to the top right, with the last 2 frames composed of blank frames.

**Figure 4 sensors-24-06975-f004:**
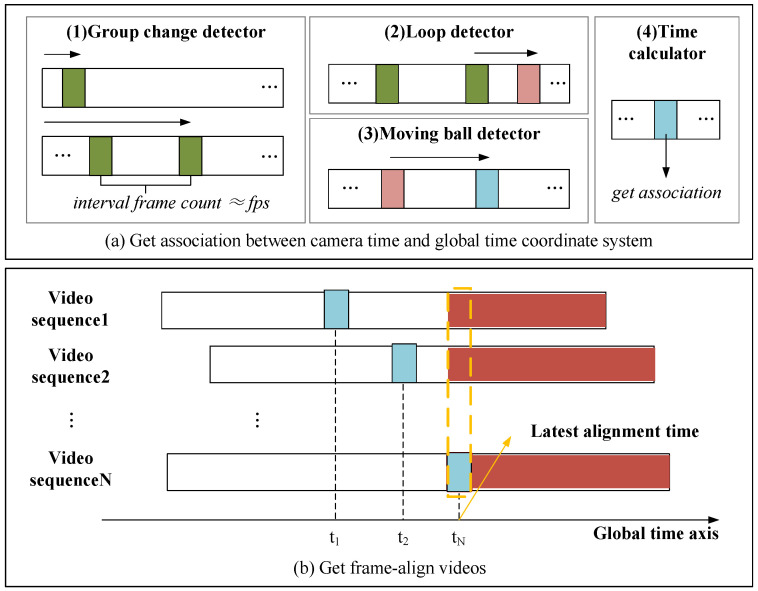
Time alignment algorithm. (**a**) The algorithm initially establishes the association between the local time systems of each camera and the global time system by detecting group information, cycle information, and moving ball information from the time-calibrated video. (**b**) Once all local time systems are associated through this global time system, the frame-aligned videos are obtained by identifying the alignment time under the global time system through these associations.

**Figure 5 sensors-24-06975-f005:**
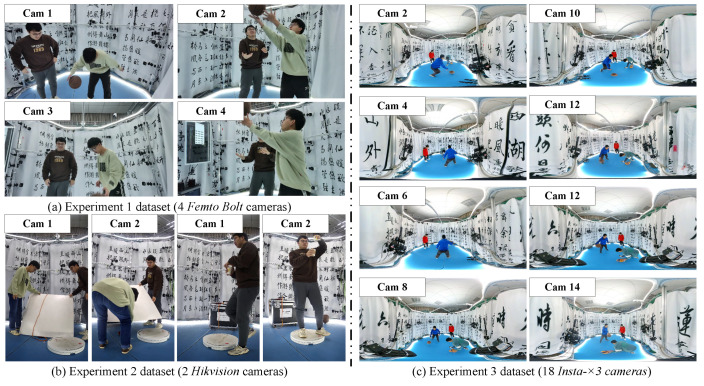
Experimental datasets sample. (**a**) Data captured by hardware-synchronous cameras (*Femto Bolt* [47]) used in the first experiment (4 cameras); (**b**) data captured by non-synchronous cameras (*Hikvision* [48]) used in the second experiment (2 cameras); (**c**) data captured by the non-synchronous panoramic cameras (*Insta*-×3 [49]) used in third experiment (18 cameras).

**Figure 6 sensors-24-06975-f006:**
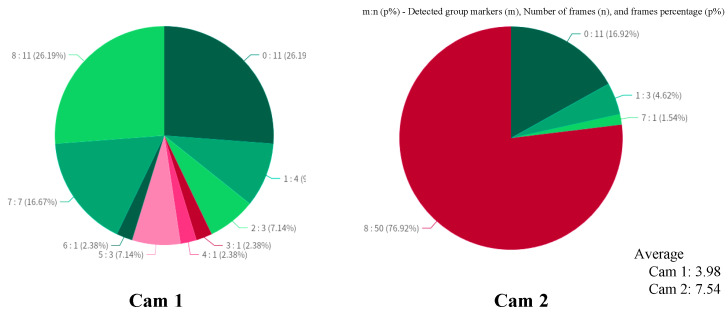
System stability validation. Due to the poor image quality captured by Cam 1, the detection stability is low, with the proportion of frames detecting only 1 marker reaching as high as 10%, and the average number of successfully detected markers being only 3.98. Cam 2 performed more stably, with an average of 7.54 markers detected per frame. This validates the rationale behind designing multiple markers per group, as it enhances detection reliability.

**Figure 7 sensors-24-06975-f007:**
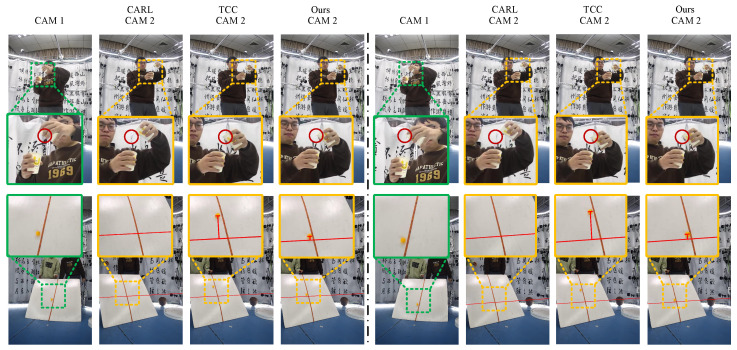
Comparison of synchronization effects under a non-synchronized camera system. There are two scenes in total, with pouring water and a falling ball. The first raw video represents pouring water and the second raw video represents the falling ball. For falling ball results, we draw the epipolar line of Cam 2. The dashed lines on the left and right sides represent consecutive frames of the video. Through visual comparison of the frames, it is evident that our algorithm yields better synchronization results compared to the other deep learning algorithms.

**Figure 8 sensors-24-06975-f008:**
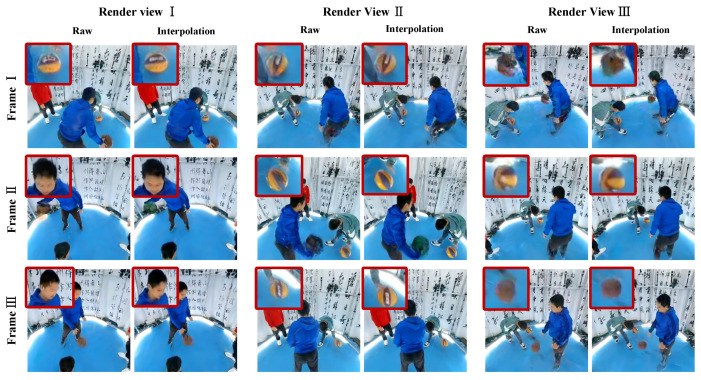
Comparison of rendering effects between non-interpolated and interpolated frames. We have selected three moments of frames to perform novel view rendering using 3D Gaussian Splatting. The results demonstrate that the rendering effects of interpolated frames are superior to those of non-interpolated frames, thus confirming that the level of synchronization has been improved after interpolation.

**Figure 9 sensors-24-06975-f009:**
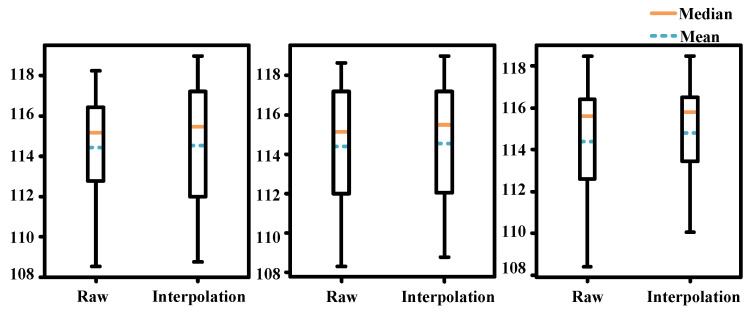
Box plots of the Laplacian gradients for rendered images using 3D Gaussian Splatting under interpolated and non-interpolated conditions. Raw represents the box plot of the rendering results for non-interpolated frames, while Interpolation represents the box plot of the rendering results for interpolated frames. There are three sets of rendered images in total. Overall, the Laplacian gradients are improved after interpolation, indicating an overall enhancement in image quality after interpolation compared to non-interpolated frames.

**Table 1 sensors-24-06975-t001:** Analysis of time performance in global time calculation.

	$t_{\mathrm{aruco}}$	$t_{\mathrm{loop}}$	$t_{\mathrm{group}}$	$t_{\mathrm{ms}}$
**Cam 1**	42.048 ms	$1.509\times 10^{-3}$ ms	$1.739\times 10^{-3}$ ms	30.459 ms
**Cam 2**	40.570 ms	$1.653\times 10^{-3}$ ms	$1.620\times 10^{-3}$ ms	33.066 ms
**Cam 3**	17.883 ms	$1.535\times 10^{-3}$ ms	$1.611\times 10^{-3}$ ms	64.950 ms
**Cam 4**	49.700 ms	$1.565\times 10^{-3}$ ms	$1.601\times 10^{-3}$ ms	32.761 ms
**Average Time**	37.550 ms	$1.566\times 10^{-3}$ ms	$1.643\times 10^{-3}$ ms	40.309 ms

**Table 2 sensors-24-06975-t002:** The error between calculated time and the real time.

Cam ID	1	2	3
0	9 ms	5 ms	12 ms
1	\	14 ms	3 ms
2	\	\	17 ms

**Table 3 sensors-24-06975-t003:** The distance calculated using epipolar geometry.

Frame ID	CARL	TCC	Ours
i−1	-	340.56 px	172.72 px
*i*	-	259.46 px	51.34 px
i+1	-	172.71 px	87.65 px

## Data Availability

The data presented in this study are available on request from the corresponding author.

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
