# Peer review of "Subframe-Level Synchronization in Multi-Camera System Using Time-Calibrated Video"

_sensors, 2024, doi:10.3390/s24216975_

Round 1

Reviewer 1 Report

Comments and Suggestions for Authors

The paper proposes a non-hardware-triggered time synchronization method based on time-calibrated video and interpolation algorithms to achieve sub-frame synchronization. This innovative approach reduces hardware dependency, lowers system complexity, and holds practical application value, especially in scenarios with numerous cameras where hardware triggering is not feasible. While the authors' idea demonstrates good innovation, the following issues need to be addressed:

1. The design of the time-calibrated video is not explained clearly enough. Please reorganize the explanation of this section in more detail.

2. It is recommended to provide more description of real-time application scenarios, analyzing the algorithm's complexity or conducting multiple experiments to verify its feasibility in real-time applications.

3. Provide a more detailed description of the experimental results.

4. The paper uses interpolation algorithms to achieve sub-frame synchronization; please explain the rationale for this choice.

5. Review all formulas in the paper and explain all variables used to improve understanding.

6. Ensure consistency in the citation format throughout the references.

7. For the self-collected dataset, provide more detailed explanations regarding the shooting environment, equipment, and other relevant details.

Comments on the Quality of English Language

English expression needs to be reviewed.

Reviewer 2 Report

Comments and Suggestions for Authors

In this work, the authors propose a sub-frame synchronization method that does not use on additional hardware triggers. Based on time-calibrated video featuring unique markers and a steadily moving object to precisely establish the temporal alignment between the local and global time references across multiple cameras.

1. The authors state in the abstract and other sections of their paper that 'Traditional methods often rely on hardware triggers or external signals, which can introduce complexities and limitations.' However, this is not entirely accurate, as technological advancements have enabled the development of highly precise and robust hardware solutions that can perform such tasks without difficulties or limitations. It is challenging to guarantee similar accuracy using methods like the one proposed in this paper, which relies on time-calibrated video with unique markers and a steadily moving object to align local and global time references across multiple cameras.

2. As the authors have claimed from the beginning that this approach offers advantages over hardware methods, can we expect a comparison between the proposed approach and a hardware-synchronized system?

3. Does the proposed approach not introduce a certain dependency on the visual markers and the moving object? Indeed, the approach relies on specific markers and a uniformly moving object for temporal calibration. However, this raises concerns about potential issues if the markers or the moving object are not clearly visible under certain conditions (e.g., poor lighting, obstructions, rapid or unpredictable movements). The method may also be limited to scenarios where the use of these elements is feasible. In dynamic, complex, or large-scale environments, where uniformly moving objects or specific markers cannot be employed, this approach may lose its effectiveness. Can you give us more details on these points and justify them?

4. While interpolation is used to refine synchronization at a sub-frame level, it is based on estimations and approximations. This may not be sufficiently accurate in applications where absolute synchronization at a very fine temporal scale is critical (such as in computer vision or critical systems). Interpolation could also introduce errors if temporal variations are not linear or uniform across cameras, potentially leading to partial desynchronization.

5. The approach seems to heavily depend on precise temporal calibration, but it does not discuss how errors in this initial calibration would affect synchronization. If the temporal calibration is slightly inaccurate, it could undermine the overall precision that the method aims to achieve.

6. I would like more details on the possible problems of this approach in terms of Computational complexity. Indeed, the recalculation of timestamps and the application of video interpolation algorithms are computationally expensive steps. These operations become particularly demanding when the system has to process large volumes of data from multiple cameras or handle high frame rates. This complexity can limit the applicability of the method in real-time environments, where latency must be minimal for quick decision-making.

7. What about these possible problems on the quality of the results of the proposed approach: Image Quality as the Resolution, and Sharpness. Effects of lossy video coding. Lighting Conditions and Contrast Variations. Occlusions and Shadows. Presence of Similar Objects in the Scene ...etc.?

8. With regard to subsection 3.1, Can you unequivocally justify the choice of ArUco markers as well as the number considered for this approach?

9. I find it somewhat surprising to read in subsection 4.1.1 that 'there are no readily available datasets for us to use directly.' While I understand that the specific requirements of time-calibrated video might make it challenging to find a perfect match, I am doubtful that there are no such databases at all. Perhaps the issue lies more in the difficulty of adapting existing datasets to the proposed approach, rather than their complete absence...!

10. From a formatting perspective, it is recommended not to place Figures 7 and 8 within the conclusion. First, include these two figures in the appropriate section, followed by the conclusion.

11. Unless I’m mistaken, the conclusion does not include any proposed future work. Why?

Comments on the Quality of English Language

The paper is quite well-written as a whole and generally clear; however, the authors could simplify certain paragraphs further for improved readability.
